# Exploring barriers and facilitators to women's intention and behavior to seek treatment for distressing sexual problems

**Julia Velten**[ID]*, **Jürgen Margraf**[ID]

Mental Health Research and Treatment Center, Ruhr University Bochum, Germany

* julia.velten@rub.de

## Abstract

Many women experience distressing problems with sexual functioning, most commonly in the form of low sexual desire or arousal, difficulties reaching orgasm, or genito-pelvic pain with sexual activity. Although effective treatments are available, more than half of the women who experience distressing sexual problems do not seek professional help. Understanding help-seeking patterns, experiences with treatment providers, and barriers to treatment is crucial to address this underutilization. Examining the role of personal characteristics, sexual problem symptoms, and cognitive factors in explaining the intention to seek treatment can help identify individuals who are most reluctant to seek help. Psychological online interventions are a promising resource to increase the availability of effective treatments. Knowledge about the predictors of women's intention to use internet-delivered treatments, as well as information about personal preferences regarding their scope, can help tailor them to women's needs. To address these research questions, cross-sectional data of 800 women ($M_{age}$ = 30.49, range = 18–73) were analyzed. While many women considered clinical psychologists to be the most qualified treatment providers, gynecologists were cited as the most likely first point of contact. Among women not utilizing any treatments, many reported a preference for dealing with a sexual problem on their own as a reason not to seek help. Higher help-seeking intention was related to living in a larger city, experiencing higher sexual distress, experiencing pain or difficulties with vaginal penetration, higher self-stigma. and lower sexual assertiveness. Women who were convinced of the effectiveness of psychological online interventions and who appreciated the benefits of anonymity indicated that they were more likely to use them. Understanding what factors influence women's decisions about whether or not to seek professional help for distressing sexual problems is key to reducing the underutilization of available resources and developing treatments that meet their needs and preferences.

**Data Availability Statement:** All relevant data are within the manuscript and its Supporting Information files.

**Funding:** We acknowledge support by the Open Access Publication Funds of the Ruhr-Universität

Bochum. The funders had no role in study design, data collection and analysis, decision to publish, or preparation of the manuscript.

**Competing interests:** The authors have declared that no competing interests exist.

# Introduction

The most common sexual problems among women are a lack of sexual desire, low sexual arousal, difficulties reaching orgasm, pain with sexual activity, or difficulties with vaginal intercourse [1]. A population-based study from the UK found that about 23% of sexually active women experienced at least one sexual problem over the last year and that about 28% of these women felt fairly or very distressed by them, yielding an annual prevalence of 4% [2]. In a population-based sample of women from Germany, a 25% lifetime prevalence for sexual difficulties associated with severe personal distress was found and 18% of sexually active women reported experiencing at least one problem over the past year [3]. Cross-national studies yield high levels of sexual problems in samples throughout the world [4]. As sexual difficulties are closely related to psychological well-being [5], mental health [6] and relationship satisfaction [7], sexual issues cannot be isolated from other aspects of women's health. Sexual functioning is associated with women's mental health, especially with symptoms of depression. This relationship is most likely bidirectional and high levels of depression can be found in sexually dysfunctional samples [8].

## Seeking help for sexual problems

The high prevalence of distressing sexual problems is contrasted by a relatively low number of women seeking professional help. In the Pfizer-founded Global Study of Sexual Attitudes, Beliefs, and Behaviours (GSSAB) from 2001 to 2002, prevalence and help-seeking behavior for sexual problems were investigated in adults between 40 and 80 years of age across 29 countries. In eight Asian countries, 45% to 88% of sexually active women with frequent or periodic sexual problems did not seek help or advice for their problem, respectively [9]. In a European GSSAB-sample, help-seeking rates were somewhat higher but still a majority of women with sexual problems did not seek professional help [10]. A Spanish GSSAB-study provided more detailed information and reported that women who had experienced at least one sexual problem sought help by talking to their partner (52%), to a medical doctor (19%), to a family member or friend (14%), or looking up information in books, magazines, or using the Internet (11%). Only five percent of affected women reported talking to a psychiatrist, psychologist, or marriage counselor [11]. In a more recent study, talking to a general practitioner was still the most common source of help. Also, 26% of women indicated consulting the Internet to search information on sexual problems [12].

Consistent with these findings, a study of 4,504 patients from German university outpatient clinics for psychotherapy showed that general mental health care is underutilized by patients with sexual problems, with only 32 (0.7%) of patients treated in these clinics receiving a sexual dysfunction diagnosis [13]. Possible reasons for this underrepresentation of patients with sexual dysfunction in general clinical settings include the lack of appropriate diagnostic tools (e.g., structured interviews for sexual dysfunction) as well as shame and stigma associated with sexual problems. Another population-based study from the UK with participants from 16 to 74 years of age suggested that patterns of help-seeking behavior might have changed in recent years, at least in Northern or Western Europe, and especially among younger individuals [12]. Among women with at least one distressing sexual problem, about 40% sought help or advice for their sex life in the past year, with younger women seeking help more than twice as often than older women (49% among 16–24-year-old women vs. 21% among women 65 years and older).

However, the overall lack of help-seeking behavior among women with distressing sexual problems is remarkable as meta-analyses support the efficacy of psychological [14] and, in some cases, medical treatments [15] for a range of sexual problems. Especially for problems relating to low sexual desire and a lack of sexual arousal, psychological interventions [16, 17]

are effective to reduce symptoms and to improve sexual satisfaction. Understanding the lack of help-seeking behavior and its associated undertreatment of distressing sexual problems is key to create effective knowledge dissemination campaigns [18] and to tailor existing treatments to women's needs and preferences.

## Barriers and facilitators for help-seeking

Results of the GSSAB-project showed that both structural (e.g., lack of access to medical care) and attitudinal (e.g., embarrassment) barriers are relevant to explain the lack of help-seeking behavior for sexual problems [9, 10]. Further, among the women who did not seek help, thinking that the problem was normal, time-constraints, and the assumption that health-care professionals could not help with sexual problems were relevant structural barriers [19]. The following sections describe the role of personal characteristics, sexual problem symptoms, and cognitive factors in determining women's intention to seek help for sexual problems.

Sexual problems affect women across age groups, sexual orientations [20], and socioeconomic statuses (SES) [21]. However, personal characteristics may play a large role in the decision to seek help for these problems, as, for example, older people may view sexual problems as largely irrelevant in the context of health care [22]. SES, defined as "the position that an individual or family occupies with reference to the prevailing average of standards of cultural possessions, effective income, material possessions, and participation in group activity in the community" [23], has been investigated in the context of seeking professional help for sexual problems. In a population-based sample from the UK, women with sexual problems who left school with a degree were more likely to have sought help for their concern as compared to women without passing exams (42% vs. 28%). In contrast, other indicators of SES (i.e., current occupation, or level of socioeconomic deprivation) were not related to help-seeking behavior [12].

The specific symptoms of sexual problems can also influence the likelihood of seeking help. In a sample of 300 women from Ghana, those experiencing pain with intercourse were more likely, and women experiencing low sexual satisfaction were less likely to seek professional help [19]. In a French GSSAB-sample, experiencing problems with lubrication was associated with more help-seeking behavior [24]. These findings are in line with a large population-based study among Australians showing that both pain during intercourse as well as lubrication problems were associated with more help-seeking behavior [25]. Overall, there appears to be evidence that sexual problems affecting vaginal intercourse, and thus may have a stronger impact on sexual behavior in heterosexual couples, result in more help-seeking compared with sexual problems affecting women's personal sexual experience (e.g., low desire, lack of mental sexual arousal, or difficulty achieving orgasm).

The way women think and feel about their sexuality, sexual problems, and help-seeking impacts their decision to seek help for distressing sexual problems. Self-stigma, defined as the diminution of a person's self-esteem or self-worth due to the perception that they are socially unacceptable [26], has been identified as a barrier to seeking psychological help for mental health problems. In two small samples of US and Swedish college students, self-stigma was associated with lower intention to seek professional help for sexual problems [27]. In a large sample of Iranian women with epilepsy, self-stigma was associated with both help-seeking intention and behavior for sexual problems [28]. The relationship between higher self-stigma and lower intention to seek help for sexual problems was mediated by lower perceived behavioral control and more negative attitudes towards seeking help from a mental health professional [29].

Sexual self-concept is a multidimensional construct and refers to a person's beliefs, attitudes, and feelings about their sexuality, including their sexual desires, preferences, and behaviors [30]. It is an important aspect of a person's overall self-concept and can influence their

sexual experiences and relationships. Two facets of the sexual self-concept, sexual assertiveness —defined as a person's tendency to be decisive and self-reliant about their sexuality [30]—and sexual self-esteem—the tendency to positively evaluate one's own capacity to engage in healthy sexual behaviors and to experience a satisfying sex life [30]—have been associated with sexual satisfaction and sexual functioning [31]. While aspects of the sexual self-concept have been investigated with respect to their relevance to sexual functioning outcomes [32], studies on the relevance for sexual help-seeking behavior are scarce.

### Current study

In the absence of in-depth data on facilitators and barriers to seeking help for sexual problems, this study will present a mix of descriptive and regression-based correlational data to examine specific themes and patterns from which falsifiable hypotheses can be derived [33] with the goal of stimulating further research and ultimately improving the care situation for affected women.

The first objective of this cross-sectional survey was to provide descriptive information about help-seeking behavior for distressing sexual problems, the perceived usefulness of different treatments, and information about barriers to seeking treatment in a large convenience sample of women in Germany. To shed more light on the different patterns of help-seeking, we also investigated how qualified different treatment providers were perceived to be and which treatment providers were considered as first points of contact.

The second objective was to examine the relevance of a broad range of predictors to women's intention to seek help for distressing sexual problems. To this end, we examined whether personal characteristics (e.g., age, partnership status, SES, sexual orientation), different symptoms of sexual problems (e.g., low desire, low arousal, difficulties reaching orgasm, or problems with vaginal penetration), or cognitive factors (e.g., self-stigma, sexual self-concept) would explain the intention to seek help for sexual problems. To this end, two regression analyses were conducted to examine predictors of help-seeking in general and of seeking psychological help.

The third objective was to examine women's attitudes toward and preferences for online psychological interventions for sexual problems. For that purpose, a regression analysis was conducted that included specific attitudes toward online interventions as potential predictors. Further, descriptive data on women's preferences for the format (e.g., self-help, standardized feedback, individualized feedback) as well as the scope (e.g., number of treatment modules, time for home exercises) of the interventions are presented.

## Materials and methods

### Participants

Participants for the online survey were recruited through multiple channels (e.g., email listservs, postings on university webpage, social media, online discussion boards) to increase sample diversity. No eligibility criteria were defined other than that participants should be cis- or trans-women and 18 years of age or older. Of the 800 participants ($M_{age}$ = 30.49, range = 18 to 73) whose data were analyzed for this study, 11 (1.4%) identified as trans or intersex. Four-hundred ninety-five (61.9%) participants reported being in a committed partnership or married, 63 (7.9%) were in an open non-monogamous or polyamorous relationship, 238 (29.7%) indicated being single, and 4 (0.5%) entered a different relationship status. Concerning sexual orientation, 368 (46.0%) and 285 (35.6%) of participants were exclusively or predominantly attracted to men, 46 (5.6%) were attracted equally to men and women, 43 (5.4%) reported that their sexual attraction did not depend on the gender of the person, 29 (3.6%) and 22 (2.8%) reported being predominantly or exclusively attracted to women, and 7 (0.9%) endorsed an

asexual orientation. A total of 537 (67.1%) of participants living in a major city with a population greater than 100,000. The majority of participants ($n$ = 650, 81.3%) reported no family history of migration, while 5.0% ($n$ = 40) reported being born outside of Germany, and 13.8% ($n$ = 100) reported having a parent or grandparent that migrated to Germany. Table 1 shows an overview of sociodemographic and socioeconomic variables.

## Measures

**Personal characteristics.** Several personal characteristics were investigated concerning their relevance for the intention to seek help for a sexual problem. The presence of a steady,

**Table 1. Sociodemographic and socioeconomic variables ($N$ = 800).**

| Variables | | $n$ (%) |
|---|---|---|
| School education | | |
| | No high-school degree | 51 (6.4) |
| | High-school degree | 749 (93.6) |
| Education | | |
| | Occupational training | 124 (15.5) |
| | Undergraduate degree | 186 (23.3) |
| | Graduate degree | 218 (27.3) |
| | Postgraduate degree | 22 (2.8) |
| | Other (e.g., student, no degree) | 250 (31.3) |
| Occupation | | |
| | Freelancer, civil servant, executive employee | 58 (7.2) |
| | Self-employed | 27 (3.4) |
| | Intermediate civil servant, farmer, qualified employee with authority to issue directives | 50 (6.3) |
| | Skilled worker, qualified employee | 147 (18.4) |
| | Lower-level civil servant | 6 (0.8) |
| | Employee without authority to issue directives | 62 (7.8) |
| | Other (e.g., blue-collar worker, student, apprentice, retiree, on parental or maternity leave) | 446 (55.8) |
| Household income (in Euro, monthly) | | |
| | Less than 750 | 82 (10.3) |
| | 750 to 1000 | 83 (10.4) |
| | 1000 to 1250 | 41 (5.1) |
| | 1250 to 1500 | 32 (4.0) |
| | 1500 to 2000 | 64 (8.0) |
| | 2000 to 2500 | 80 (10.0) |
| | 2500 to 3000 | 65 (8.1) |
| | 3000 to 3500 | 59 (7.4) |
| | 3500 and more | 143 (17.9) |
| | I don't know | 76 (9.5) |
| | I choose not to tell | 75 (9.4) |
| Socioeconomic status (self-reported) | | |
| | Lowest class | 14 (1.8) |
| | Working class | 39 (4.9) |
| | Lower middle class | 116 (14.5) |
| | Middle class | 395 (49.4) |
| | Upper middle class | 223 (27.9) |
| | Upper class | 13 (1.6) |

monogamous partnership was included as a dichotomous predictor with being single and other relationship forms (e.g., consensual non-monogamy) as being coded "no". To examine the influence of urbanicity, participants were asked about the size of their current place of residence, with cities more than 100,000 people classified as "urban" and all other areas (e.g., villages, small towns) as "rural". Sexual orientation was also used as a dichotomous variable with exclusively heterosexual orientation as one category and all other orientations as another. To calculate the objective SES, three variables were combined: Occupation and education level of the main income earner, and net household income. To identify the main income earner, participants were asked to indicate whether they themselves or another person contributed more to the household income [34]. As these objective variables may not be appropriate to estimate the socioeconomic situation of certain populations, e.g., university students, we also assessed their subjective SES by asking participants to identify as members of the lower, working, lower middle, middle, upper middle, or upper social class. Information concerning the socioeconomic characteristics of the main income earner of the household, used to calculate an objective indicator of SES, are presented in the S1 Table.

**Depression and anxiety.** The Patient Health Questionnaire-4 (PHQ-4) [35] is an ultra-short self-report scale that assesses symptoms of anxiety and depression over the last two weeks with 4 items ranging from 0 (not at all) to 3 (nearly every day). Internal consistency of this measure was good ($\alpha$ = .87) in the present sample.

**Sexual problems.** The Sexual Complaints Screener for Women (SCS-W) is a screening tool estimating women's sexual problems with sexual functioning over the past 6 months [36]. The questionnaire assesses the frequency and distress associated with low sexual desire (2 items), a lack of sexual arousal (4 items), problems reaching orgasm (2 items), problems with genito-pelvic pain or penetration (4 items) on items rated on a 5-point Likert-scale ranging from 0 (never) to 4 (almost all the time/always) for the frequency items and 0 (not at all a problem) to 4 (a very great problem) for the distress items. Additional SCS-W items for persistent genital arousal as well as sexual satisfaction were not included in this study. Convergent validity of the German version of the SCS-W was acceptable to excellent for the different domains [36]. In this study, the internal consistency for the subscales ranged from $\alpha$ = .69 for low desire to $\alpha$ = .85 for genito-pelvic pain/problems with penetration ($\alpha$ = .85 for the complete scale).

**Sexual distress.** Sexuality-related personal distress during the past 30 days was assessed using the validated Sexual Distress Scale–Short Form (SDS-SF) [37]. The five items were rated using a 5-point Likert-scale ranging from 0 (never) to 4 (always) and a total score ranging from 0 to 20 with higher scores indicating greater sexual distress. The SDS-SF has demonstrated excellent reliability and discriminant validity [37]. In this study, internal consistency was good ($\alpha$ = .86).

**Help-seeking intention.** The Mental Help Seeking Intention Scale (MHSIS) [38] is a 3-item questionnaire measuring respondents' intention to seek help from a mental health professional if they had a mental health concern. Items are rated on a 6-point Likert scale from 1 to 7 with higher scores indicating greater intention. The MHSIS has shown good predictive validity by correctly predicting, with almost 70% accuracy, future help-seeking behavior of adults with a current mental health concern [38]. For the purpose of this study, items were modified in order to assess help-seeking intention for sexual problems specifically (e.g., "*If I had a sexual problem, I would try to seek help from a mental health professional*"). Further, two additional versions–inquiring about the intention to seek help in general (e.g., "*If I had a sexual problem, I would try to seek help.*") and the intention to utilize a psychological online-intervention (e.g., "*If I had a sexual problem, I would try to seek help in the form of a psychological online-intervention.*") were administered.

**Sexual self-concept.**  Four subscales of the Multidimensional Sexual Self-Concept Questionnaire (MSSCQ) [39] were used to assess feelings and attitudes related to a person's sexuality: Sexual self-efficacy (e.g., *"I am competent enough to make sure that my sexual needs are fulfilled."*) describes the belief that a person has the ability to effectively deal with sexual aspects of their life, Sexual problem management (e.g., *"If I were to experience a sexual problem, I myself would be in control of whether this improved."*) describes the belief that a person has the skills to effectively handle any sexual problems, Sexual assertiveness (e.g., *"I'm very assertive about the sexual aspects of my life."*) describes a person's tendency to be decisive and self-reliant about their sexuality, and Sexual Self-esteem (e.g., *"I derive a sense of self-pride from the way I handle my own sexual needs and desires."*) describes a person's tendency to positively evaluate their own capacity to engage in healthy sexual behaviors and to experience a satisfying sex life. Each subscale consisted of 5 items rated on a 5-point scale that ranged from 1 (not at all characteristic of me) to 5 (very characteristic of me). The construct validity of the MSSCQ has been shown (Snell, 2001) and internal consistency of the subscales ranged between $\alpha = .84$ for Sexual problem management and $\alpha = .92$ for Sexual self-esteem in the present study.

**Structural and attitudinal barriers.**  Perceived barriers to seeking treatment for distressing sexual problems were assessed with a modified 14-item checklist developed as part of the World Mental Health surveys [40]. This checklist includes six structural barriers (e.g., concerns about costs, not getting an appointment) and eight attitudinal barriers (e.g., expecting the problem to get better by itself, lack of trust in treatment effectiveness). Participants were instructed to select all items they perceived as relevant for their lack of help-seeking behavior for sexual problems.

**Self-stigma of seeking help.**  The Self-Stigma of Seeking Help scale (SSOSH) [41] assesses the degree to which individuals may avoid seeking psychological help to preserve autonomy and self-worth. The scale consists of ten items rated on a 5-point Likert-scale ranging from 1 (strongly disagree) to 5 (strongly agree). Self-stigma is thought to occur when people experiencing a mental disorder or considering psychological help self-label as socially unacceptable and thereby internalize stereotypes [42]. The SSOSH has shown good psychometric properties and measurement invariance across countries [41, 43]. In this study, a modified version of the SSOSH was used to assess women's willingness to seek psychological help for sexual problems, specifically (e.g., *"If I went to a therapist for a sexual problem, I would be less satisfied with myself."*).

**Treatments for sexual problems.**  To explore women's experiences with different kinds of treatments for sexual problems, a checklist of eleven potential treatments was presented. The list was created based on previous studies on help-seeking behavior and included different health care professions (e.g., psychologist, psychiatrists, physicians, gynecologists, or naturopaths) as well as different settings (e.g., outpatient treatments, inpatient treatments, counselling centers, psychological online interventions). Further, participants could indicate whether they received couples counselling, or received prescription or over-the-counter medications for their sexual problem. For each treatment utilized, participants then indicated on a 5-point scale from 1 (not at all helpful) to 5 (very helpful) how helpful they experienced it in addressing their sexual problem.

All participants were also asked which health care providers they considered most qualified to handle a potential sexual problem and which provider they would seek out first if they had a problem. To answer these questions, a list of eight professions was presented, and participants were asked to rank them in the order most appropriate for them. To score these rankings, the first choice in each case was given 8 points, the 2nd choice was given 7 points, and so on. Professions that were not selected by the participants received 0 points.

**Attitudes towards and preferences for psychological online interventions.** The Attitudes towards Psychological Online Interventions (APOI) scale is a 16-item measure assessing cognitively based attitudes towards such interventions [44]. Items are rated on a 5-point scale from 1 (do not agree at all) to 5 (agree completely) and factor analysis revealed four lower order factors. Scepticism and perception of risks (4 items, e.g., *"I do not expect any longer-term effects from online psychological interventions.")* describes peoples' doubts about long-term effectiveness as well as potential negative outcomes such as increased loneliness, Confidence in effectiveness (4 items, e.g., "*I feel that an online psychological intervention could help me.*") describes trust and confidence in the helpfulness of online interventions, Technologization threat (4 items, e.g., *"In a crisis situation*, *a therapist can help me better than an online psychological intervention."*) describe benefits of an in-person therapist as compared to an online intervention, and Anonymity benefits (4 items, e.g., *"An online psychological intervention is more confidential and discreet than sex therapy with a therapist."* describes how online interventions can be more discreet and may make it easier to reveal personal feelings as compared to seeing a therapist. In this study, a modified version of the APOI specifically asking about attitudes towards psychological online interventions for sexual problems was used. Internal consistency of the subscales ranged from $\alpha$ = .65 (Anonymity benefits) to $\alpha$ = .83 (Confidence in effectiveness).

To evaluate preferences for psychological online interventions, participants were asked to rate the following versions: Self-help only (no feedback by study personnel), Supported self-help (feedback by study personnel only on request), Standardized feedback on completed modules, Individual feedback on completed modules by trained support personnel, and Individual feedback on completed modules by psychologist and/or sex therapist on a 5-point scale from 1 (not at all helpful/appealing) to 5 (very helpful/appealing). Further, participants were asked about the preferred number of treatment modules as well as the optimal duration of each module (in minutes). Psychological at-home exercises to be completed between modules (or therapy sessions) are a crucial part of the treatment for sexual dysfunctions, thus, participants were asked how many minutes on how many days a week they would prefer to set aside for at-home exercises to be done alone and with a sexual partner. In order to get an overall estimate of the amount of time participants would spend on working on a sexual problem, they were asked to estimate the total number of hours they would be willing and able to spend.

## Procedure

On the first page of the survey, visitors were informed about the sensitive content of the study. Those who decided to proceed, provided written informed consent online, and were forwarded to the questionnaire. After a first set of sociodemographic questions, the SCS-W assessing the frequency and distress associated with different sexual problems, was presented. Women who reported having experienced at least one sexual problem sometimes, often, or (almost) always during the last 6 months, and who felt that this constituted some problem, a considerable problem, or a very great problem for them were considered as experiencing a distressing sexual problem and were forwarded to a page that asked about which of their problems, they perceived as most distressing.

All participants were then asked about their help-seeking intention in case they experienced sexual problems. Then, they were asked whether they had sought out any treatment for problems with sexual functioning in the past. If they responded affirmatively, they were presented with a list of potential treatments and were asked first to indicate if they ever had ever sought treatment for a sexual problem and then to rate each treatment on its perceived helpfulness.

To explore the relevance of structural and attitudinal barriers to seeking help for sexual problems, women who indicated experiencing distressing sexual problems over the last six months, and who never utilized any treatments for sexual problems, were first asked whether the reason for not seeking help was that the problem went away on its own. Women who negated this were then presented a list of potential barriers for treatment [40] and were asked to check all barriers they perceived as being relevant for their decision not to seek treatment. The remaining questionnaire including measures of e.g., self-stigma, sexual self-concept, symptoms of depression and anxiety, as well as socioeconomic variables was then presented to all participants. On the last page of the survey, participants could opt-in to enter their email-address for a prize draw or receive course credit for participation. The study was carried out in accordance with the provisions of the World Medical Association Declaration of Helsinki (2013). The Ethical Committee of the Faculty of Psychology at the Ruhr University Bochum formally approved the study including the procedure to assess and document informed consent (Reference No. 73).

## Data analysis

Data were analyzed using SPSS 29 (SPSS, Inc, Chicago, IL, USA). Descriptive values (i.e., means, medians, standard deviations, and percentages) were reported for the experiences with treatments, treatment barriers, and the evaluation of sexual health care providers. To investigate predictors of help-seeking intention for sexual problems, zero-order correlations between predictors and outcomes (i.e., intention to seek help in general, to seek psychological help, and to utilize psychological online interventions) were calculated. In total, four multiple regression analyses were conducted to examine the relation between predictor and outcome variables.

## Results

### Descriptive analyses

Out of 800 participants, 150 women (18.8%) reported at least one distressing sexual problem in the past six months. Among women with distressing sexual problems, 44 (29.3%) indicated being most bothered by problems reaching orgasm, 33 (22.0%) by low sexual arousal, 33 (22.0%) by pain or problems relating to vaginal penetration, and 18 (12.0%) by low sexual desire. In the complete sample, levels of sexual distress as measured with the SDS-SF were 5.15 ($SD$ = 4.20) with 256 women (32.0%) scoring above a clinical cut-off score of 7 [37]. Help-seeking intention for sexual problems measured with the MHSIS was 4.05 ($SD$ = 1.73) for seeking help in general, 3.41 ($SD$ = 1.78) for seeking psychological help, and 3.04 ($SD$ = 1.75) for utilizing psychological online interventions, implying that on average, participants were undecided whether to seek help or not for a sexual problem. Mean levels of depression and anxiety as measured by the PHQ-4 were 4.09 ($SD$ = 3.04) suggesting that on average, participants exhibited mild symptoms of psychopathology. Mean values of self-stigma as measured with the SSOSH were 24.32 ($SD$ = 7.70). Using four subscales of the MSSCQ yielded mean levels of 2.44 ($SD$ = 0.85) for Sexual self-efficacy, 1.99 ($SD$ = 0.95) for Sexual assertiveness, 2.51 ($SD$ = 0.70) for Sexual problem management, and 2.14 ($SD$ = 0.98) for Sexual self-esteem.

**Experiences with treatments for sexual problems.** About half of the participants reported ($n$ = 382, 47.7%) that they had sought treatment for a sexual problem in the past. By far the most frequently mentioned treatment was consultation with a gynecologist, followed by over-the-counter medications, and couples counselling. On average, most of the utilized treatments were rated as somewhat helpful, with lowest ratings found for consultations by a general practitioner (see Table 2).

Table 2. Experience with treatments for sexual difficulties (*N* = 800).

| | Endorsement, n (%), multiple answers possible | Satisfaction with treatment, 1 (not helpful at all) to 5 (very helpful) |
|---|---|---|
| | *n* (%) | *M(SD)* |
| No treatment at all | 418 (52.3) | - |
| Consultation(s) with gynecologist | 240 (30.0) | 3.19 (1.27) |
| Over-the-counter medications | 58 (7.2) | 3.15 (1.30) |
| Couples counseling or therapy | 37 (4.6) | 3.58 (1.21) |
| Prescription medications | 23 (2.9) | 3.20 (1.32) |
| Psychotherapy, outpatient setting | 22 (2.8) | 3.58 (1.35) |
| Consultation(s) at (sexual) counseling center (e.g., planned parenthood) | 21 (2.6) | 3.47 (1.26) |
| Consultation(s) with general practitioner | 18 (2.3) | 2.88 (1.31) |
| Consultation(s) with psychiatrist | 13 (1.6) | 3.91 (0.83) |
| Psychological online intervention | 13 (1.6) | 3.11 (0.93) |
| Other treatment | 13 (1.6) | 3.82 (0.98) |
| Consultation(s) with naturopath | 14 (1.8) | 3.50 (1.17) |
| Psychotherapy, inpatient setting | 3 (0.4) | 4.00 (1.41) |

**Barriers to treatment.** To explore reasons for treatment non-utilization, women who indicated distressing sexual problems in the past six months, and who did seek out any treatment despite the problem persisting were asked about potential structural and attitudinal barriers (see Table 3).

Table 3. Treatment barriers for women with sexual distressing sexual problems (*n* = 150).

| Why did you not seek treatment for your sexual problem(s)? | Endorsement, *n* (%) |
|---|---|
| Low perceived need: The problem went away by itself, and I did not really need help. | 61 (40.7) |
| Other reasons (see below) | 89 (59.3) |
| **Structural barriers** | **Endorsement, *n* (%), multiple answers possible** |
| I was unsure about where to go or who to see. | 50 (56.8) |
| I thought it would take too much time or be inconvenient. | 49 (55.7) |
| My health insurance would not cover this type of treatment. | 15 (17.9) |
| I was concerned about how much money it would cost. | 13 (15.5) |
| I had problems with things like transportation, childcare, or scheduling that would have made it hard to get to treatment. | 5 (5.9) |
| I could not get an appointment. | 2 (2.4) |
| **Attitudinal barriers** | |
| I wanted to handle the problem on my own. | 77 (87.5) |
| I thought the problem would get better by itself. | 66 (75.0) |
| The problem didn't bother me very much. | 46 (52.3) |
| I didn't think treatment would work. | 35 (40.7) |
| I was concerned about what others might think if they found out I was in treatment. | 25 (29.4) |
| I was not satisfied with available services. | 8 (9.5) |
| I received treatment before and it did not work. | 5 (6.0) |
| I was scared about being put into a hospital against my will. | 3 (3.5) |

Despite indicating the presence of at least one distressing sexual problem in the past six months, 40.9% (*n* = 61) reported low perceived need in the sense that the problem went away on its own. The remaining 89 women (59.3%) were presented with a checklist of potential barriers. Among the structural barriers, "*I was unsure about where to go or who to see.*" and "*I thought it would take too much time or be inconvenient.*" were endorsed by more than 50% of the women. Only a minority of women indicated reasons relating to health insurance, costs, or transportation, or difficulties getting an appointment. Among the attitudinal barriers, "*I wanted to handle the problem on my own.*" and "*I thought the problem would get better by itself.*"were endorsed by 75% or more women. About half of the women agreed to the statement "*The problem didn't bother me very much*" despite indicating the presence of at least one distressing sexual problem in the SCS-W.

**Perception of sexual health care providers.** To explore the perception of different sexual health care providers, participants were asked to rank providers from most to least qualified for treating sexual problems and from most to least likely first point of contact (see Table 4).

Clinical psychologists, gynecologists, as well as (sexual) counseling centers (e.g., planned parenthood) were most frequently listed as the most qualified treatment providers for sexual problems. These three providers were also the most likely to be the first point of contact, with gynecologists being the most likely first contact.

## Intention to seek help for sexual problems

Table 5 shows bivariate correlations between predictor and outcome variables. Although some of the variables did not show significant bivariate correlations with help-seeking intentions, all variables were included in the regression models.

Table 6 shows the results of the regression analysis for general help-seeking intention for sexual problems.

Two personal characteristics added to the explanation of this intention, as women who lived in major cities as well as those who had received prior care for sexual problems showed higher intention to seek help. Concerning sexual functioning, low desire and arousal problems were associated with lower intention, difficulties or pain with vaginal penetration were associated with higher intention. Two cognitive factors also added to the explanation, self-stigma for seeking help was associated with lower intention and sexual assertiveness—the tendency to be decisive and self-reliant about sexual matters—was associated with higher intention. The

**Table 4. Evaluation of sexual health care providers (*N* = 800).**

|  | Most qualified (0 "not qualified" to 8 "most qualified) | First point of contact (0 "would not contact" to 8 "most likely to contact first") |
|---|---|---|
|  | *M (SD)* | *M (SD)* |
| Clinical psychologist (i.e., therapist with degree in psychology) | 6.06 (2.49) | 4.63 (3.32) |
| Gynecologist | 5.05 (2.91) | 5.34 (3.36) |
| (Sexual) counseling center | 4.76 (2.91) | 3.63 (3.36) |
| Couple counselor | 4.68 (2.71) | 3.19 (3.21) |
| Medical psychotherapist (i.e., therapist with medical degree) | 3.43 (2.92) | 1.88 (2.70) |
| Psychiatrist, Neurologist | 1.74 (2.28) | 0.94 (1.87) |
| General practitioner | 1.07 (1.73) | 1.27 (2.29) |
| Naturopath | 0.96 (1.76) | 0.79 (1.80) |

**Table 5. Speakman rank correlations between predictor and outcome variables.**

| | | 1 | 2 | 3 | 4 | 5 | 6 | 7 | 8 | 9 | 10 | 11 | 12 | 13 | 14 | 15 | 16 | 17 | 18 | 19 | 20 |
|---|---|---|---|---|---|---|---|---|---|---|---|---|---|---|---|---|---|---|---|---|---|
| 1 | Help-seeking intention | -- | | | | | | | | | | | | | | | | | | | |
| 2 | Help-seeking intention–psychological | .67*** | -- | | | | | | | | | | | | | | | | | | |
| 3 | Help-seeking intention–online | .23*** | .35*** | -- | | | | | | | | | | | | | | | | | |
| 4 | Age | .16*** | .20*** | -.02 | -- | | | | | | | | | | | | | | | | |
| 5 | Partnership (yes/no) | .02 | -.02 | -.01 | .09* | -- | | | | | | | | | | | | | | | |
| 6 | Urbanicity | .08* | .11** | .09** | .00 | .00 | -- | | | | | | | | | | | | | | |
| 7 | Sexual orientation | -.08* | -.07* | -.02 | .04 | .13*** | -.04 | -- | | | | | | | | | | | | | |
| 8 | Objective socioeconomic status | .07* | .05 | -.03 | .32*** | .16*** | -.01 | .05 | -- | | | | | | | | | | | | |
| 9 | Subjective socioeconomic status | .00 | -.07 | .01 | -.10** | .09** | .04 | .08* | .25*** | -- | | | | | | | | | | | |
| 10 | Depression and anxiety | -.10** | -.05 | -.07 | -.13** | -.11** | .01 | -.13*** | -.11** | -.13*** | -- | | | | | | | | | | |
| 11 | Any prior help for sexual concerns | .20*** | .10** | .02 | .06 | .01 | -.01 | -.14*** | .02 | -.04 | .13*** | -- | | | | | | | | | |
| 12 | Sexual distress | -.15*** | -.03 | .01 | -.02 | .05 | .00 | -.09* | -.02 | -.11** | .37*** | .22*** | -- | | | | | | | | |
| 13 | Low sexual desire | -.18*** | -.08* | -.02 | .01 | .10** | .06 | -.08* | .01 | -.04 | .20*** | .16*** | .51*** | -- | | | | | | | |
| 14 | Lack of arousal | -.20** | -.11** | -.05 | -.06 | .08* | -.01 | -.05 | -.05 | -.04 | .19*** | .18*** | .43*** | .53*** | -- | | | | | | |
| 15 | Difficulties reaching orgasm | -.17*** | -.13*** | -.03 | -.20*** | -.02 | -.02 | -.02 | -.05 | -.05 | .11** | .11** | .30*** | .22*** | .39*** | -- | | | | | |
| 16 | Painful Intercourse | -.07 | -.13*** | -.03 | -.20*** | .02 | -.01 | -.05 | -.05 | .01 | .26*** | .27*** | .33*** | .25*** | .35*** | .27*** | -- | | | | |
| 17 | Self-stigma for seeking help | -.44*** | -.42*** | -.12** | -.20*** | -.01 | -.08* | .03 | .02 | -.06 | .18*** | -.06 | .17*** | .08* | .13** | .09* | .11** | -- | | | |
| 18 | Sexual self-efficacy | .25*** | .12** | .02 | .06 | .03 | .02 | -.03 | .02 | .03 | -.24*** | -.10** | -.43*** | -.34*** | -.40** | -.29*** | -.22*** | -.27*** | -- | | |
| 19 | Sexual assertiveness | .34*** | .19*** | .03 | .08* | .09* | .00 | .00 | .03 | .03 | -.19*** | .01 | -.33*** | -.27*** | -.25** | -.25*** | -.21*** | -.27*** | .60*** | -- | |
| 20 | Sexual problem management | .12** | .11** | .01 | .11** | -.03 | .03 | .03 | .02 | -.02 | -.17*** | -.06 | -.21*** | -.12** | -.17** | -.13*** | -.16*** | -.08* | .36*** | .21*** | -- |
| 21 | Sexual esteem | .32*** | .17*** | .02 | .07* | -.01 | .04 | .04 | .04 | .03 | -.26*** | -.07 | -.46*** | -.38*** | -.37*** | -.30*** | -.26*** | -.31*** | .80*** | .74*** | .29*** |

*** = $p < .001$;

** = $p < .01$;

* = $p < .05$

**Table 6. Multiple regression analysis to predict help-seeking intention for sexual problems.**

| | b | SE | β | t | p | 95.0% CI Lower | 95.0% CI Upper |
|---|---|---|---|---|---|---|---|
| Constant | 5.11 | .61 | | 8.40 | < .001 | 3.91 | 6.30 |
| Personal characteristics | | | | | | | |
| Age | .01 | .01 | .05 | 1.34 | .182 | .00 | .02 |
| Partnership (yes/no) | .04 | .11 | .01 | 0.36 | .721 | -.18 | .27 |
| Urbanicity (rural/urban) | .25 | .11 | .07 | 2.26 | .024 | .03 | .48 |
| Sexual orientation (other/excl. heterosexual) | -.17 | .11 | -.05 | -1.50 | .134 | -.38 | .05 |
| Objective socioeconomic status | .16 | .09 | .06 | 1.71 | .088 | -.02 | .34 |
| Subjective socioeconomic status | -.10 | .06 | -.05 | -1.59 | .112 | -.22 | .02 |
| Depression and anxiety (PHQ-4) | .00 | .02 | -.01 | -0.16 | .872 | -.04 | .04 |
| Any prior help for sexual problems | .59 | .12 | .17 | 5.01 | < .001 | .36 | .83 |
| Sexual problem symptoms | | | | | | | |
| Sexual distress (SDS-SF) | .03 | .02 | .07 | 1.69 | .091 | .00 | .06 |
| Low desire (SCS-W) | -.17 | .07 | -.10 | -2.56 | .011 | -.30 | -.04 |
| Lack of arousal (SCS-W) | -.19 | .09 | -.10 | -2.21 | .027 | -.36 | -.02 |
| Difficulties reaching Orgasm (SCS-W) | -.08 | .05 | -.06 | -1.58 | .114 | -.19 | .02 |
| Painful Intercourse (SCS-W) | .15 | .07 | .08 | 2.21 | .028 | .02 | .29 |
| Cognitive factors | | | | | | | |
| Self-stigma for seeking help (SSOSH-S) | -.08 | .01 | -.36 | -10.64 | < .001 | -.09 | -.07 |
| Sexual self-efficacy (MSSCQ) | -.10 | .11 | -.05 | -0.93 | .353 | -.32 | .12 |
| Sexual assertiveness (MSSCQ) | .29 | .08 | .16 | 3.50 | < .001 | .13 | .46 |
| Sexual problem management (MSSCQ) | .11 | .08 | .05 | 1.36 | .174 | -.05 | .27 |
| Sexual esteem (MSSCQ) | .12 | .11 | .07 | 1.10 | .270 | -.10 | .34 |

MSSCQ = Multidimensional Sexual Self-Concept Questionnaire. PHQ-4 = Patient Health Questionnaire 4, SCS-W = Sexual Complaints Screener for Women, SDS-SF = Sexual Distress Scale Short Form. SSOSH-S = Self-Stigma of Seeking Help Scale–Sexual

model explained about 31.3% of outcomes variance. Table 7 shows the results of the regression analysis for the intention to seek psychological help for sexual problems.

Higher age, living in a larger city, and having a lower subjective SES were associated with higher intention. Concerning sexual functioning, experiencing more sexuality-related distress was associated with higher, the presence of orgasm problems with lower intention. Among the cognitive factors, only higher sexual assertiveness was associated with higher intention. The model for the explanation of the intention to seek psychological help explained a total of 25.9% of variance.

## Psychological online interventions

First, we used the same regression model to explore predictors of the intention to seek help for sexual problems in the form of psychological online interventions. However, this model explained only 4.9% of outcome variance (see S2 Table for the complete model), with living in a larger city and low self-stigma as the only significant predictors. Thus, a second model including a measure of attitudes towards psychological online interventions was calculated. Adding this measure to the model increased the amount of variance explained to 29.4%. Higher levels on the APOI scales Confidence in effectiveness and Anonymity benefits were associated with higher intention to utilize psychological online interventions.

**Table 7. Multiple regression analysis to predict the intention to seek psychological help for sexual problems.**

| | b | SE | β | t | p | 95.0% CI Lower | 95.0% CI Upper |
|---|---|---|---|---|---|---|---|
| Constant | 5.33 | .65 | | 8.16 | < .001 | 4.05 | 6.61 |
| **Personal characteristics** | | | | | | | |
| Age | .01 | .01 | .07 | 2.06 | .040 | .00 | .02 |
| Partnership (yes/no) | -.10 | .12 | -.03 | -0.78 | .434 | -.34 | .15 |
| Urbanicity (rural/urban) | .33 | .12 | .09 | 2.68 | .008 | .09 | .56 |
| Sexual orientation (other/excl. heterosexual) | -.13 | .12 | -.04 | -1.06 | .289 | -.36 | .11 |
| Objective socioeconomic status | .13 | .10 | .05 | 1.28 | .201 | -.07 | .32 |
| Subjective socioeconomic status | -.19 | .07 | -.10 | -2.85 | .005 | -.33 | -.06 |
| Depression and anxiety (PHQ-4) | .03 | .02 | .05 | 1.33 | .185 | -.01 | .07 |
| Any prior help for sexual problems | .33 | .13 | .09 | 2.57 | .010 | .08 | .58 |
| **Sexual problem symptoms** | | | | | | | |
| Sexual distress (SDS-SF) | .05 | .02 | .11 | 2.37 | .018 | .01 | .08 |
| Low desire (SCS-W) | -.13 | .07 | -.08 | -1.86 | .063 | -.28 | .01 |
| Lack of arousal (SCS-W) | -.04 | .09 | -.02 | -0.44 | .661 | -.22 | .14 |
| Difficulties reaching Orgasm (SCS-W) | -.12 | .06 | -.08 | -2.13 | .034 | -.23 | -.01 |
| Painful Intercourse (SCS-W) | -.12 | .08 | -.06 | -1.60 | .109 | -.27 | .03 |
| **Cognitive factors** | | | | | | | |
| Self-stigma for seeking help (SSOSH-S) | -.12 | .12 | -.06 | -1.03 | .304 | -.36 | .11 |
| Sexual self-efficacy (MSSCQ) | .17 | .09 | .09 | 1.84 | .066 | -.01 | .34 |
| Sexual assertiveness (MSSCQ) | .25 | .09 | .10 | 2.87 | .004 | .08 | .43 |
| Sexual problem management (MSSCQ) | -.03 | .12 | -.02 | -0.29 | .773 | -.27 | .20 |
| Sexual esteem (MSSCQ) | -.12 | .12 | -.06 | -1.03 | .304 | -.36 | .11 |

MSSCQ = Multidimensional Sexual Self-Concept Questionnaire. PHQ-4 = Patient Health Questionnaire 4, SCS-W = Sexual Complaints Screener for Women, SDS-SF = Sexual Distress Scale Short Form. SSOSH-S = Self-Stigma of Seeking Help Scale–Sexual

On average, women found self-help only interventions ($M = 2.52$, $SD = 1.11$), and interventions with standardized feedback ($M = 2.85$, $SD = 1.09$) the least appealing, followed by supported self-help ($M = 3.34$, $SD = 1.06$). The two versions including individual feedback by trained support personnel ($M = 4.08$, $SD = 0.93$) or by a psychologist and/or sex therapist ($M = 4.56$, $SD = 0.84$) were rated as most appealing.

Women's preferences in terms of the scope of the interventions, yielded a total of 9.57 ($Median = 7$, $SD = 20.68$) modules with an average length of 34.98 minutes ($Median = 30$, $SD = 19.51$) as the optimal length. Concerning at-home exercises, women indicated a willingness to spend about 17.83 minutes ($Median = 15$, $SD = 15.64$) on 3.17 ($Median = 3$, $SD = 1.56$) days a week with individual exercises, and about 20.57 minutes ($Median = 20$, $SD = 15.82$) on 2.08 ($Median = 2$, $SD = 2.26$) days a week with partner exercises. In total, women estimated the total amount of time they would be able to spend working on a sexual problem at 3.84 hours ($Median = 2$, $SD = 11.21$) per week.

## Discussion

This study investigated barriers and facilitators to women's intention and behavior to seek treatment for distressing sexual problems. First, the perception of different care providers was reported, followed by the role of structural and attitudinal barriers in women's decisions not to seek treatment. Then, the role of personal characteristics, symptoms of sexual problems, and

cognitive factors for the explanation of help-seeking intention was analyzed. Finally, women's intention to utilize psychological online interventions as well as their personal preferences for such treatments were explored.

## Barriers and facilitators to help-seeking behavior

About half of women in this study reported some prior treatment utilization. This finding is in line with recent studies suggesting an overall uptake in help-seeking behavior, especially among younger women [12]. Most commonly, women had consulted their gynecologist for advice on sexual problems. This reflects a common perception as medical doctors being considered first line and psychological interventions second line treatments [45]. Also, this finding is not surprising as annual gynecological cancer screenings [46] provide a good opportunity for women to address sexual concerns without the need to make a separate appointment. Women also endorsed the use of over-the-counter medications with the goal to improve their sexual problems. While these may include nutritional supplements without any proven benefits for sexual functioning, they might also include lubricants or cremes to improve a lack of vaginal lubrication during sex. As expected, utilization of more intensive psychological treatments was reported by a minority of women [13].

Women who reported a distressing sexual problem in the past six months and indicated that they had not sought treatment were first asked if the problem had gone away on its own. Women who responded negatively were presented a checklist of structural and attitudinal barriers to treatment. Two structural barriers were endorsed by more than half of the women: Many did not know where or to whom to go with their sexual problem, and they thought it would take too much time or be uncomfortable. The first barrier can be overcome by disseminating information about qualified sexual health care providers to the public through social media campaigns [18] and by making information about the effectiveness of psychological treatments available to other stakeholders, such as gynecologists or general practitioners. The second structural barrier, however, may be more difficult to overcome as effective psychological treatments do require a considerable personal commitment of time and energy. Communicating this upfront may help women determine whether they are willing and able to pursue treatment under their current circumstances. For nearly 90% of women, a preference for self-management of the problem was another relevant attitudinal barrier. As today's society places a great burden and responsibility on women to be an active member of the workforce as well as taking care of their home, children, and extended family, many women may find it difficult for them to devote time to their own care. For these women, effective self-help-based psychological treatments–either via bibliotherapy [47] or in the form of psychological online interventions [17]–can be suitable options. In addition, it would be important to convey that other psychological therapies for sexual problems are also essentially self-help, and that the main part of the treatment consists of completing sex therapy exercises between sessions [48, 49].

## Explanation of help-seeking intention

Previous help-seeking behavior was a significant predictor of help-seeking intention both for help in general and for psychological help, more specifically. Among the personal characteristics, only urbanicity added to the explanation in that women living in larger cities had higher help-seeking intention. Since Germany is a highly developed and densely populated country and even small towns and rural areas have relatively good access to health care, the urbanicity factor may reflect a perception of more readily available treatment providers as sexual health services may be more present in urban areas through advertisements, practice signs, and

counseling centers. When other variables were controlled for, neither age nor sexual orientation, depression/anxiety explained the intention to seek help.

Women with higher subjective SES indicated lower intention to seek psychological help and objective SES was not predictive of help-seeking intention. Previous studies showed that the relationship between SES and help-seeking for sexual problems may be more complex as compared to, e.g., other mental health issues where individuals with lower SES seem to show lower rates of help-seeking behavior [50]. In a 2019 population based-study from the UK, objective SES as measured by two indices was not associated with help-seeking, however, women who left high-school before the age of 16 without passing exams showed lower rates of help-seeking behavior [12]. Thus, one reason for the null finding could be that women without high-school diploma or those who may be part of marginalized communities or in particularly difficult life situations (e.g., women seeking asylum or living in shelters) did not participate in this current study.

With regard to the role of sexual problem symptoms, the findings are consistent with previous studies which have suggested that higher levels of sexuality-related personal distress as well as difficulties or pain with vaginal penetration are associated with higher help-seeking, whereas problems related to a lack of sexual desire are associated with lower help-seeking intention [25]. Informing women about their right to pleasure can strengthen their sense of entitlement to a satisfying sex life. This may encourage them to seek help for e.g., low desire sexual problems even if they did not experience difficulties in engaging in vaginal intercourse. Combating stereotypes about female sexuality, emphasizing the role of personal pleasure, and disseminating information on psychological treatments that are especially effective in improving low sexual desire might improve their confidence to seek treatment [51, 52].

Among the cognitive factors, self-stigma was associated with lower general help-seeking intention and sexual assertiveness with higher intention to seek both psychological and general help. Two types of interventions to reduce self-stigma have shown promise. First, interventions attempting to alter stigmatizing beliefs and attitudes directly as well as interventions aiming to enhance skills for coping with self-stigma through improvements in self-esteem and empowerment [53]. Fostering women's sexual assertiveness and sexual empowerment [54] is important to allow women to take agency over their own sexuality, prioritize their own pleasure, seek help in instances where they feel distressed by a sexual problem, but also to allow them to behave less sexually compliant and potentially reduce sexual victimization [55]. Overall, the two models explaining general and psychological help-seeking intention yielded similar results in that there was substantial overlap in the relevant predictors and that the complete models explained a comparable amount of outcome variance.

## Psychological online interventions

An additional model explaining the intention to utilize psychological online interventions for sexual problems explained only a fraction of outcome variance suggesting that variables not included in the model were more relevant in explaining this outcome. In fact, specific attitudes toward such programs were highly relevant, with women who felt confident in the effectiveness of such interventions reporting higher intention. A 2022 meta-analysis on psychological online interventions yielded mixed effects for sexual dysfunctions in men. Results for women, however, were promising with large effects comparable to those found in face-to-face treatments [17]. Communicating these findings to the public could therefore be a means of increasing confidence and hence uptake. In addition, the lack of direct face-to-face contact, which some women may perceive as a disadvantage of internet-delivered treatments, may be an

advantage for others who may feel more comfortable expressing intimate aspects of their sexual lives more openly in this anonymous setting.

Overall, women's ambivalence towards such interventions was reflected in lower levels of intention as compared to other treatment options which may be indicative of a polarized public debate about the implementation of e-mental health services [56]. Studies have found rather low awareness of the availability and underutilization of such interventions [12]. However, since the presentation of neutral, psychoeducational information led to some increases in the intention to use internet-based interventions, using the media, for example, to disseminate this information may be a means of increasing women's willingness to use them [56]. Descriptive information on women's treatment preferences can guide future intervention development. Ratings of the appeal of different degrees of support revealed women's preference for personal, individualized feedback as compared to self-help only interventions. Regular standardized feedback, however, was perceived as less appealing as supported self-help which may include personalized feedback to submitted questions. Such personalized feedback, even if only available on request, might help to establish a therapeutic alliance between participants and the person providing the feedback as well as the online-intervention itself [57, 58]. Preferences concerning the scope of the interventions showed that women, on average, preferred interventions with about 10 modules of about 35 minutes length. These preferences match some of the recently validated interventions for low sexual desire and genito-pelvic pain [59–61]. Concerning the scope of at-home exercises, however, the preferred 17 to 20 minutes on about two or three days a week are somewhat lower than what is usually expected from participants in psychological online interventions for sexual problems. These preferences for somewhat shorter practice durations and frequencies may be driven by the (mis)assumption that sex therapy is primarily based on verbal interventions [45]. Most psychological online interventions, however, are based on either a cognitive-behavioral or mindfulness-based framework and go beyond simply providing verbal (i.e., text-based) information but also include at-home exercises specific for the respective therapeutic approach (i.e., filling out thought diaries or practicing mindfulness) in addition to sex-therapy specific exercises (e.g., self-exploration, sensate focus). Based on this study, researchers and clinicians administering psychological online interventions should communicate the amount of recommended at home-exercises beforehand to set the right expectations and increase motivation and adherence.

## Limitations

This study used a convenience sample of relatively young and highly educated women. The sample may have been representative in terms of the prevalence of sexual problems [3] and participants' sexual orientations but findings cannot be directly generalized to older and less educated populations. While the descriptive analysis on prior help-seeking assessed actual treatment utilization, our regression models focused on the intention to seek help. While help-seeking intention is a strong predictor of the behavior [62], it cannot substitute the assessment of actual behaviors [63]. Longitudinal studies are thus needed to disentangle the role of personal variables, sexual function symptoms, and cognitive factors in the intention-behavior relationship. Further, relationship characteristics such as emotional intimacy [64] as well as other partner-related factors (e.g., a partner's sexual satisfaction) [7] can impact sexual distress as well as help-seeking intention, thus, future studies should employ a dyadic approach to account for these interpersonal factors. As help-seeking intention is largely determined by the quality and availability of health-care services, studies allowing for comparisons across countries are needed to determine the relevance of individual vs. societal factors. Lastly,

investigating physical health variables as predictors would be worthwhile to determine the role of overall health in the explanation of help-seeking intention.

## Conclusion

Although effective treatments for sexual problems in women are available, more than half of the women who experience distressing sexual problems do not seek professional help. Understanding help-seeking patterns, experiences with treatment providers, and barriers to treatment is crucial to address this underutilization. This study emphasized the role of urban vs. rural living areas, sexual problem symptoms, as well as self-stigma and sexual assertiveness in explaining the intention to seek treatment. Further, confidence in the effectiveness and a perception of anonymity benefits was associated with women's intention to utilize psychological online interventions. Understanding factors influencing women's decisions about whether or not to seek professional help for distressing sexual problems is key to reducing the underutilization of available resources and developing effective treatments that meet women's needs and preferences.

## Supporting information

**S1 Table. Sociodemographic and socioeconomic variables for the main income earner ($N$ = 800).**
(DOC)

**S2 Table. Explanation of intention to utilize psychological online interventions for sexual problem.**
(DOC)

**S1 Dataset.**
(SAV)

## Author Contributions

**Conceptualization:** Julia Velten, Jürgen Margraf.

**Data curation:** Julia Velten.

**Formal analysis:** Julia Velten.

**Investigation:** Julia Velten.

**Methodology:** Julia Velten.

**Project administration:** Julia Velten.

**Resources:** Jürgen Margraf.

**Supervision:** Jürgen Margraf.

**Validation:** Julia Velten.

**Writing – original draft:** Julia Velten, Jürgen Margraf.

**Writing – review & editing:** Julia Velten, Jürgen Margraf.

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
