## [Decision Letter · Decision Letter 0]

26 Apr 2023

PONE-D-23-05351Exploring barriers and facilitators to women's intention and behavior to seek treatment for distressing sexual problemsPLOS ONE

Dear Dr. Velten,

Thank you for submitting your manuscript to PLOS ONE. After careful consideration, we feel that it has merit but does not fully meet PLOS ONE’s publication criteria as it currently stands. Therefore, we invite you to submit a revised version of the manuscript that addresses the points raised during the review process.

The paper is really well written and exaustive. Please answer minor points risen by reviewers. According to me the method section is complete.

We look forward to receiving your revised manuscript.

Kind regards,

Marta Panzeri, Ph.D.

Academic Editor

PLOS ONE

Journal Requirements:

2.  Please provide additional details regarding participant consent. In the ethics statement in the Methods and online submission information, please ensure that you have specified (1) whether consent was informed and (2) what type you obtained (for instance, written or verbal, and if verbal, how it was documented and witnessed). If your study included minors, state whether you obtained consent from parents or guardians. If the need for consent was waived by the ethics committee, please include this information. If you are reporting a retrospective study of medical records or archived samples, please ensure that you have discussed whether all data were fully anonymized before you accessed them and/or whether the IRB or ethics committee waived the requirement for informed consent. If patients provided informed written consent to have data from their medical records used in research, please include this information.

Reviewers' comments:

Reviewer's Responses to Questions

**Comments to the Author**

1. Is the manuscript technically sound, and do the data support the conclusions?

Reviewer #1: Yes

Reviewer #2: Yes

2. Has the statistical analysis been performed appropriately and rigorously? 

Reviewer #1: Yes

Reviewer #2: Yes

3. Have the authors made all data underlying the findings in their manuscript fully available?

Reviewer #1: Yes

Reviewer #2: Yes

4. Is the manuscript presented in an intelligible fashion and written in standard English?

Reviewer #1: Yes

Reviewer #2: Yes

5. Review Comments to the Author

Reviewer #1: The paper is rich and nicely written. It tackled the topic of help seeking for sexual problems from different perspectives and provided further understanding of the factors that affect women’s intention and behaviour.

I would suggest considering the following points:

Introduction:

Elaborate the impact of sexual problems on women’s life and wellbeing.

The context of the study should be elaborated highlighting the healthcare system and the facilities provided for sexual healthcare services. This was not mention knowing that the accessibility, availability, quality, and acceptability of care could be major facilitators or barriers to helpseeking for sexual problems. Another factor that impacts on women’s helpseeking behaviour is the husband attitude which also should be reflected on.

Methods:

The design is not mentioned.

Since data collection was done online, I wonder if data about the country of residence and nationality were collected as the culture may affect help seeking.

How did you ensure consistency of the definition of the sociodemographic and socioeconomic variables among participants coming from different countries?

Reviewer #2: Dear Editor and Authors:

This manuscript aims to identify barriers and facilitators that lead women to seek treatment for sexual problems.

It is a complete, complex and well elaborated work. In addition, the effort made by the authors to compose it can be observed, since they use a large number of variables and manage to defend them in a very adequate way. This study expands the knowledge about the behavioral intention to seek help that women present and can contribute both in clinical practice and as a starting point for future research and interventions.

Page 3 – Introduction: To complete the introduction, it is recommended to include some data on sexual difficulties worldwide.

Page 5 - Barriers and facilitators for help-seeking: This section is very complete. The first paragraph talks about structural and attitudinal characteristics. It is recommended to order the information in this section, e.g. following the content of the first paragraph, i.e. structural and attitudinal characteristics or according to the variables to be used in this study. In this line, it is recommended to mention these characteristics. For example, at the beginning of the paragraphs that are developed in this section, in order to order the information and make it easier to read for future users of the journal. In addition to socio-economic status, it is recommended that any further structural barriers be included or made clearer in the text.

It is considered necessary to include a justification for the inclusion of the variables under study and not others.

Page 8 Participants: Can the authors specify whether there were any exclusion criteria?

Pages 9-14 Measures: It is recommended to include examples of items from the scales used.

Page 11 – Line 145 “professional.”)”: Include a full stop at the end of the sentence.

Page 11 - Help-seeking intention: Can the authors explain why they only refer to mental health? Please include examples of items from the two additional versions indicated in the paragraph.

Page 12 Self-stigma of seeking help/ Attitudes towards and preferences for psychological online-interventions: Can the authors explain why no such questionnaire on physical health was included? This should be clear from the introduction.

Page 16, Lines 370-377: It is recommended to indicate in summary form what these data, these averages, mean. This makes it easier for the reader to understand quickly.

Table 3 “Other reasons”: Please indicate in brackets or at the bottom of the table what "other reasons" mean.

Lines 482-500: Today's society places a greater burden and responsibility on women, which can make it difficult for them to devote time to their own care. There may be a need for health education for the whole population, with a special emphasis on women. Perhaps those women who report having no time are working in and out of the home, taking care of the family, etc. In future studies it would be interesting to learn about these aspects that may act as barriers to general and sexual health care for these women.

Lines 521-522 “It is also conceivable that women who are in a particularly good socioeconomic 522 situation fear a loss of status to a greater extent if they initiate treatment for a sexual problem”: Please elaborate in the text on this conclusion reached by the authors (and cite).

Were people who indicated any sexual problems or sexual distress offered any help or benefits for participating in this study (other than those mentioned in the Method)?

Journal checklist:

1. The study presents the results of original research. X

2. Results reported have not been published elsewhere.

3. Experiments, statistics, and other analyses are performed to a high technical standard and are described in sufficient detail. X

4. Conclusions are presented in an appropriate fashion and are supported by the data.X

5. The article is presented in an intelligible fashion and is written in standard English.X

6. The research meets all applicable standards for the ethics of experimentation and research integrity. X

7. The article adheres to appropriate reporting guidelines and community standards for data availability.X

6. PLOS authors have the option to publish the peer review history of their article (what does this mean?). If published, this will include your full peer review and any attached files.

Reviewer #1: **Yes: **Mathilde Azar

Reviewer #2: No

---

## [Author Response · Author response to Decision Letter 0]

10 May 2023

Journal Requirements:

a. We have checked the files and ensured that they meet the requirements. 

2. Please provide additional details regarding participant consent. In the ethics statement in the Methods and online submission information, please ensure that you have specified (1) whether consent was informed and (2) what type you obtained (for instance, written or verbal, and if verbal, how it was documented and witnessed). 

a. We have verified that this information is in the manuscript. 

a. We have added the reference number to the manuscript. 

a. Checked. 

Reviewer #1: 

1. The paper is rich and nicely written. It tackled the topic of help seeking for sexual problems from different perspectives and provided further understanding of the factors that affect women’s intention and behaviour. I would suggest considering the following points:

• We would like to thank this reviewer for their appreciation of our manuscript. We hope to have addressed all their comments to their satisfaction. 

2. Introduction:Elaborate the impact of sexual problems on women’s life and wellbeing.

• To address this comment, we have added information on the bidirectional relationship between mental health problems such as depression and women’s sexual functioning to the introduction. 

3. The context of the study should be elaborated highlighting the healthcare system and the facilities provided for sexual healthcare services. This was not mention knowing that the accessibility, availability, quality, and acceptability of care could be major facilitators or barriers to helpseeking for sexual problems. 

• Thanks for this very helpful suggestion, we now discuss the relevance of the availability and quality of health-care services in the context of multi-national studies.

4. Another factor that impacts on women’s helpseeking behaviour is the husband attitude which also should be reflected on.

• We agree with this comment and have added a section on interpersonal factors to the discussion section. As we have not included such variables in our analysis, we decided not to emphasize them in the introduction. 

5. Methods: The design is not mentioned.

• We now describe the study as a cross-sectional survey in the last section of the introduction. 

6. Since data collection was done online, I wonder if data about the country of residence and nationality were collected as the culture may affect help seeking. How did you ensure consistency of the definition of the sociodemographic and socioeconomic variables among participants coming from different countries?

• More than 80% of participants reported no family history of migration in that they and their parents/grandparents were born in Germany. Although we did not ask for this information, it is reasonable to assume that the vast majority of participants were taking the survey from inside of Germany. Also, migration status had no significant association with help-seeking intention. 

Reviewer #2: 

7. This manuscript aims to identify barriers and facilitators that lead women to seek treatment for sexual problems.It is a complete, complex and well elaborated work. In addition, the effort made by the authors to compose it can be observed, since they use a large number of variables and manage to defend them in a very adequate way. This study expands the knowledge about the behavioral intention to seek help that women present and can contribute both in clinical practice and as a starting point for future research and interventions.

• We would like to thank this reviewer for their overall appreciation of our manuscript. We hope that we have addressed all their comments to their satisfaction. 

8. Page 3 – Introduction: To complete the introduction, it is recommended to include some data on sexual difficulties worldwide.

• We have referenced the GSSAB study to support the notion that sexual problems are very prevalent across countries world-wide. 

9. Page 5 - Barriers and facilitators for help-seeking: This section is very complete. The first paragraph talks about structural and attitudinal characteristics. It is recommended to order the information in this section, e.g. following the content of the first paragraph, i.e. structural and attitudinal characteristics or according to the variables to be used in this study. In this line, it is recommended to mention these characteristics. For example, at the beginning of the paragraphs that are developed in this section, in order to order the information and make it easier to read for future users of the journal. In addition to socio-economic status, it is recommended that any further structural barriers be included or made clearer in the text.

• We labeled the structural barriers that are listed in the first paragraph as such to improve clarity. We have structured the introduction, methods, results, and discussion section to report on personal characteristics first, followed by the role of sexual problems, as well as cognitive factors. We have also added an advanced organizer to the introduction in order to improve readability. 

10. It is considered necessary to include a justification for the inclusion of the variables under study and not others. 

• In the context of this explorative study, we decided to include a broad range of variables without relying on previous data, thus, we have decided not to include a post-hoc justification. 

11. Page 8 Participants: Can the authors specify whether there were any exclusion criteria?

• As described on Page 8, there were no exclusion criteria defined for this study (“No eligibility criteria were defined other than participants being cis- or trans-women and were to be 18 years or older”)

12. Pages 9-14 Measures: It is recommended to include examples of items from the scales used.

• Thanks for this suggestion. We have added examples of items for all relevant questionnaires. 

13. Page 11 – Line 145 “professional.”)”: Include a full stop at the end of the sentence.

• Done. 

14. Page 11 - Help-seeking intention: Can the authors explain why they only refer to mental health? Please include examples of items from the two additional versions indicated in the paragraph.

• As the original measure refers to a mental health professional, we used this phrase in one of our versions as well. Further, we inquired about general help-seeking intentions as well as using a psychological online-intervention. As suggested, we have added example items for all three versions. 

15. Page 12 Self-stigma of seeking help/ Attitudes towards and preferences for psychological online-interventions: Can the authors explain why no such questionnaire on physical health was included? This should be clear from the introduction.

• We acknowledge this as a shortcoming of our study and agree that it would be worthwhile to examine the role of physical health as well and have added this to the limitation section. 

16. Page 16, Lines 370-377: It is recommended to indicate in summary form what these data, these averages, mean. This makes it easier for the reader to understand quickly.

• We have added a short interpretation of the help-seeking and psychopathology scores in this section. 

17. Table 3 “Other reasons”: Please indicate in brackets or at the bottom of the table what "other reasons" mean.

• We have clarified that “other reasons” means that participants were presented with the list of potential reasons that is presented below in the same table. 

18. Lines 482-500: Today's society places a greater burden and responsibility on women, which can make it difficult for them to devote time to their own care. There may be a need for health education for the whole population, with a special emphasis on women. Perhaps those women who report having no time are working in and out of the home, taking care of the family, etc. In future studies it would be interesting to learn about these aspects that may act as barriers to general and sexual health care for these women.

• Thanks for raising this very interesting point. We have added a statement toward that end to the discussion section. 

19. Lines 521-522 “It is also conceivable that women who are in a particularly good socioeconomic 522 situation fear a loss of status to a greater extent if they initiate treatment for a sexual problem”: Please elaborate in the text on this conclusion reached by the authors (and cite).

• We agree that this statement would warrant a further discussion and have decided to omit it as part of the revision.

20. Were people who indicated any sexual problems or sexual distress offered any help or benefits for participating in this study (other than those mentioned in the Method)?

• No, they were not offered any help as part of the study.

---

## [Decision Letter · Decision Letter 1]

21 Jun 2023

Exploring barriers and facilitators to women's intention and behavior to seek treatment for distressing sexual problems

PONE-D-23-05351R1

Dear Dr. Velten,

We’re pleased to inform you that your manuscript has been judged scientifically suitable for publication and will be formally accepted for publication once it meets all outstanding technical requirements.

Kind regards,

Marta Panzeri, Ph.D.

Academic Editor

PLOS ONE

Additional Editor Comments (optional):

Reviewers' comments:

Reviewer's Responses to Questions

**Comments to the Author**

1. If the authors have adequately addressed your comments raised in a previous round of review and you feel that this manuscript is now acceptable for publication, you may indicate that here to bypass the “Comments to the Author” section, enter your conflict of interest statement in the “Confidential to Editor” section, and submit your "Accept" recommendation.

Reviewer #1: All comments have been addressed

2. Is the manuscript technically sound, and do the data support the conclusions?

Reviewer #1: Yes

3. Has the statistical analysis been performed appropriately and rigorously? 

Reviewer #1: Yes

4. Have the authors made all data underlying the findings in their manuscript fully available?

Reviewer #1: Yes

5. Is the manuscript presented in an intelligible fashion and written in standard English?

Reviewer #1: Yes

6. Review Comments to the Author

Reviewer #1: The authors answered all the questions and addressed all the comments. The introduction is well elaborated with emphasis on the variables of the study. The rational is clearly highlighted considering the context and the gaps in the literature. The methods are presented and clearly explained. The data are rich and support the research questions. The discussion is rich and the conclusion is relevant.

7. PLOS authors have the option to publish the peer review history of their article (what does this mean?). If published, this will include your full peer review and any attached files.

Reviewer #1: **Yes: **Mathilde Azar

---

## [Editor Report · Acceptance letter]

10 Jul 2023

PONE-D-23-05351R1 

Exploring barriers and facilitators to women's intention and behavior to seek treatment for distressing sexual problems 

Dear Dr. Velten:

I'm pleased to inform you that your manuscript has been deemed suitable for publication in PLOS ONE. Congratulations! Your manuscript is now with our production department. 

Kind regards, 

on behalf of

Dr. Marta Panzeri 

Academic Editor

PLOS ONE